# MAPPinfo - mapping quality of health information: Validation study of an assessment instrument

Jürgen Kasper[1]*, Julia Lühnen[2], Jana Hinneburg[2], Andrea Siebenhofer[3,4], Nicole Posch[3], Birte Berger-Höger[2], Alexander Grafe[5], Milada Cvancarova Småstuen[1], Anke Steckelberg[2]

1 Department of Nursing and Health Promotion, Faculty of Health Sciences, Oslo Metropolitan University, Oslo, Norway, 2 Institute of Health and Nursing Science, Faculty of Medicine, Martin Luther University, Halle (Saale), Germany, 3 Institute of General Practice and Evidence-Based Health Services Research, Medical University of Graz, Graz, Austria, 4 Institute for General Practice, Goethe University Frankfurt am Main, Frankfurt am Main, Germany, 5 MSH Medical School Hamburg, Hamburg, Germany

* jurgenka@oslomet.no

**Data Availability Statement:** All relevant data are within the paper.

**Funding:** The development was funded by the German foundation: Stiftung Gesundheitswissen.

## Abstract

### Background

Health information is a prerequisite for informed choices–decisions, made by individuals about their own health based on knowledge and in congruence with own preferences. Criteria for development, content and design have been defined in a corresponding guideline. However, no instruments exist that provide reasonably operationalised measurement items. Therefore, we drafted the checklist, MAPPinfo, addressing the existing criteria with 19 items.

### Objectives

The current study aimed to validate MAPPinfo.

### Methods

Five substudies were conducted subsequently at the Martin Luther University Halle-Wittenberg, Germany and the Medical University of Graz, Austria: (1) to determine content validity through expert reviews of the first draft, (2) to determine feasibility using 'think aloud' in piloting with untrained users, (3) to determine inter-rater reliability and criterion validity through a pretest on 50 health information materials, (4) to determine construct validity using 50 developers' self-declarations about development methods as a reference standard, (5) to determine divergent validity in comparison with the Ensuring Quality Information for Patients (EQIP) (expanded) Scale. The analyses used were qualitative methods and correlation-based methods for determining both inter-rater reliability and validity.

The funders had no role in study design, data collection and analysis, decision to publish, or preparation of the manuscript.

**Competing interests:** The authors have declared that no competing interests exist.

## Results

The instrument was considered by experts to operationalise the existing guidelines convincingly. Health and nursing science students found it easy to understand and use. It also had good interrater reliability (mean of T coefficients = .79) and provided a very good estimate of the reference standard (Spearman's rho = .89), implying sound construct validity. Finally, comparison with the EQIP instrument revealed important and distinct areas of similarities and differences.

## Conclusions

The new instrument is ready for use as a screening instrument without the need for training. According to its underpinning concept the instrument exclusively comprises items which are justified by either ethics or research evidence, implying negligence of not yet evidence based, however, potentially important criteria. Further research is needed to complete the body of evidence-based criteria, aiming at an extension of the guideline and MAPPinfo.

## Trial registration number

AsPredicted22546; date of registration: 24 July 2019.

## Background

Citizens demand for more information and in particular more reliable information as well as greater participation in the decision-making process has repeatedly been reported [1]. These two elements constitute the term "informed choice" which means that those concerned have sufficient knowledge and the decisions are in line with individual preferences [2]. Knowledge qualifying for informed choices can be achieved through evidence-based health information (EBHI). Criteria for EBHI have been defined by national and international working groups. Aiming at assuring the users' knowledge and willingness to engage in own decisions, standards for the development and design of patient decision aids have been agreed upon in an international Delphi process, resulting in the IPDAS criteria [3–5]. Also, efforts have been undertaken in research to determine evidence-based criteria on health information [6]. An evidence-based guideline was developed including a structured consensus process providing recommendations based on both ethical requirements and systematic evidence synthesis [7]. It addresses the development process, target group orientation, the content of evidence-based health information and how the information is presented, such as the presentation of frequencies or the use of pictures and graphics. However, information complying with the recommendations of the guideline is only provided for a few decisions. In contrast, the availability of low-quality health information is huge. It has been shown that it is a challenge for both citizens and health-care professionals to understand and assess health information [8].

Therefore, support judging the quality of health information is urgently needed, especially for lay people.

Therefore, the project "Mapping the quality of health information" (MAPPinfo) was initiated to operationalise the criteria of the guideline evidence-based health information, thus providing an instrument to assess the quality of health information [9].

At the beginning, we conducted a systematic literature review for available instruments that are capable of assessing the quality of health information according to the guideline criteria.

We identified 27 assessment tools. However, none of them fully fulfilled all of the four defined requirements (validated, evidence-based criteria complete, non-evidence-based criteria avoided, operationalisation provided) [9]. Eleven tools fulfilled two or three of the requirements. In addition, none of the tools can be used by untrained lay people.

## Aim of the study

In this article, we report the results of the first validation study of MAPPinfo which were gained in accordance with a previously published protocol [9]. The study determined the checklist's reliability in terms of inter-rater-agreement. The study also ascertained the extent to which MAPPinfo, which is constructed as a screening instrument working upon the immediately observable features of information material, provides an accurate estimate of the quality of health information regarding the entire set of the guideline's recommendations [7]. To allow for relating MAPPinfo to other frequently used assessment instruments, divergent validity estimates were gained with the instrument: "Ensuring quality information for patients" (EQIP [10]), which has similar structure and quality categories. It was the overarching goal of this sequence of research steps, to provide a ready-to-go measure for quality assessment of health information in any medical domain in accordance with the existing guideline.

## Methods

### Design

After conceiving and drafting the instrument, five validation steps were carried out (hereafter called sub-studies). An English translation of the validated final version of the instrument is accessible at the homepage of the Stiftung Gesundheitswissen [11]. A fact sheet on the instrument is provided in Box 1, an overview with items and domains in Table 1. The project has been approved by the ethics committee of the Martin Luther University Halle-Wittenberg (approval number: 2019 115).

### Substudy 1: Expert review / content validity

The instrument passed expert reviews in the first step, thus assuring content validity.

**Sample.** Two experts (MA and SB) for evidence-based health information, not previously involved in the development, were appropriately recruited via email from the Network Evidence-based Medicine (EbM) and gave informed consent.

**Data collection.** The draft was sent to the experts who were asked to comment on distinctiveness and exhaustivity of the set of criteria as well as on whether the criteria were correctly defined with regard to the current state of knowledge on evidence-based health information.

**Analysis.** The research group discussed comments and objections and guided the initial revision.

### Substudy 2: Pilot testing (trial application)

To test and optimise the checklist's feasibility, MAPPinfo was under participating observation tried out by representatives of the target group.

**Samples.** Students of health and nursing science from the Faculty of Medicine at the Martin Luther University Halle-Wittenberg were appropriately recruited and gave informed consent. Initially two students piloted the instrument. New students were recruited successively to test revisions. We chose test information materials from two domains (one from each), contraception and gonarthrosis. Websites were randomly chosen from a comprehensive pool of health information in these domains, which had been gained by systematic Google searches.

## Box 1. Overview MAPPinfo checklist

| MAPPinfo checklist fact sheet | |
|---|---|
| Scope of application | Material (regardless of media) developed to inform (users in making) health-related decisions implying a choice between at least two alternativesNot applicable to:<br>• information about the structure of health services,<br>• informed consent documentation to legally hedge the health care provider,<br>• describing the conduct of a medical procedure after a decision already has been made |
| Subject of evaluation | Whether and to which extent health information material is facilitate informed choices |
| Source of quality criteria | a. Ethical guidelines [12]<br>b. Evidence [7] |
| Type of instrument | MAPPinfo is a screening instrument (estimating the quality based on a set of easily observable criteria) that works as a checklist. |
| Selection of criteria | • If either evidence or ethics imply a clearly directed recommendation<br>• and the criterion is observable without inclusion of secondary sources |
| Users | People with basic knowledge of evidence-based medicine but without any training as an evaluator |
| Domains (number of items) | Definitions (2)<br>Transparency (6)<br>Content (7)<br>Presentation (8) |
| Limits of appraisal | Methods of development, evidence update, user involvement and evaluation |
| Administration | Items are presented with either dichotomous (0, 1) or trichotomous (0, 0.5, 1) answer format. A manual provides a definition and explanation in plain language for each item and a best practice example |

**Data collection.** In a Zoom meeting, an interviewer (JK) provided the participants with the checklist and manual. Then they were asked to conduct an appraisal of the test materials using the think aloud method [13]. The interviewer worked as both moderator and observer of the appraisal process. When necessary, the interviewer encouraged the participants to share thoughts and considerations to disclose as much information as possible about comprehensibility and feasibility of the items. The moderator also used observations made during the process to sharpen the focus on potential inconsistencies or barriers.

**Analysis.** Obvious problems identified during the test applications were immediately discussed in the research group and a solution was figured out which was then tested in a new interview. Solutions could refer to the general instructions, the manual, the wording of the items or the answering format. After all the problems were solved, a new version, revision 2 was finalised.

## Substudy 3: Pretest

The third substudy was to determine the new instrument's reliability in terms of inter-rater-reliability and therefore employed a pretest, where a larger amount of health information material was appraised by several raters. In addition, a first estimate of validity was to be gained using an expert rating as a reference.

**Samples.** Two new students from the same programme (see substudy 2) were recruited and 50 websites randomly chosen from the comprehensive pools of health information materials on contraception and gonarthrosis (25 of each domain). The sample size was determined

**Table 1. Overview of items of the MAPPinfo checklist.**

| Domain, itemno. | Item | Assessment |
|---|---|---|
| Definitions | | |
| 1 | The target group addressed by the health information is clearly defined. | 0 / 1 |
| 2 | The health information explains explicitly that an informed choice about a concrete problem should be facilitated. | 0 / 0,5 / 1 |
| Transparency | | |
| 3 | The authors of the health information are named. | 0 / 0,5 / 1 |
| 4 | The funding source of the health information is disclosed. | 0 / 1 |
| 5 | A strategy for managing conflicts of interest is disclosed. | 0 / 1 |
| 6 | The health information indicates how up-to-date it is. | 0 / 0,5 / 1 |
| 7 | The sources of information are named. | 0 / 0,5 / 1 |
| 8 | The systematic search strategies underlying the generation of information are transparent. | 0 / 1 |
| Content | | |
| 9 | The health problem is explained. | 0 / 1 |
| 10 | Options are named and explained. | 0 / 0,5 / 1 |
| 11 | The health information makes statements about stochastic uncertainty. | 0 / 1 |
| Content / Presentation | | |
| 12 /16 | The natural course (in the case of diagnostic problems: the prevalence) of the disease is adequately presented. | 0 / 1 |
| 13 / 17 | The benefit is presented adequately. | 0 / 1 |
| 14 / 18 | The harm is presented adequately. | 0 / 1 |
| 15 / 19 | For diagnostic problems: Information on the quality of the test is presented appropriately. | 0 / 1 |
| Presentation | | |
| 20 | The health information uses a neutral language throughout. | 0 / 1 |
| 21 | The health information does not use narratives that present relevant factual information. | 0 / 1 |
| 22 | Where applicable, graphics are designed in a suitable manner. | 0 / 0,5 / 1 |
| 23 | Information about benefits / harm are supplemented with complementary information (Gain Loss Framing). | 0 / 0,5 / 1 |

balancing the workload and the wish to provide a solid database. Two members of the research group (AS, JK) worked as an expert team.

**Data collection.** The two students and the expert team independently rated all 50 websites. The students used the MAPPinfo checklist and documented their assessments of the MAPPinfo items in two separate Excel files. Then the two student raters discussed all disagreements and documented a third row of assessments, representing consensus. The expert team used a 10-point rating scale (0 = very low quality of the entire information, 10 = excellent quality of the entire information) documenting their assessments in a fourth data column.

**Analyses.** T-coefficients were calculated between the two raters and Spearman's coefficients between the consensus-mean value and the expert rating. T is a modified Cohen's kappa coefficient using theoretic assumptions rather than empiric data to calculate expected frequencies [14, 15]. This decision reflects the assumption that the rater should make judgements on the MAPPinfo checklist on a theoretical basis regardless of the distribution of quality criteria in the specific information sample. T- and Spearman's coefficients were considered moderate when between 0.40 and 0.60, strong when higher than 0.60, and excellent when higher than 0.80 [16]. Insufficient reliability (lower than 0.4) on item level was considered an indication of

the need for another qualitative investigation of the particular items (see substudy 2) and a repetition of the pretest with newly recruited raters afterwards. This sequence was continued until T-coefficients of all items were at least moderate.

## Substudy 4: Comparison with self-declarations by health information developers

As the most important validation step within the agenda of the current study, construct validation was determined using a cross-sectional design by comparing MAPPinfo with a reference standard. In the absence of a gold standard, we created an extended scale, the MAPPinfo-plus by adding quality criteria obtained from self-declarations by health information developers.

**Sample.** To allow for maximal variation regarding style, interests, financing, and level of quality, we contacted developers of health information from nine different groups: health insurances, confessional bodies, patient organisations, doctors' surgeries, public organisations, hospital trusts, specialist organisations, foundations and commercial providers. It was important to identify persons involved in the development processes of health information. Fifty developers were included and asked to fill in an online questionnaire providing details about the development process of a certain health information, which they also were asked to specify. As only descriptive statistics were planned and no information on the distribution of data was available apriori, the decision on the size of the sample was made in due consideration of the practical challenges of recruitment. Informed consent was included in the online questionnaire.

**Data collection.** Using LimeSurvey, an online questionnaire was developed by the research team to obtain insight into the methods used during the development process and thereby completing the basis of assessment with regard to the quality criteria suggested by the guideline [8]. In particular, the questionnaire asked for methods used to 1. identify the specific needs of the target group, 2. identify relevant evidence, 3. consider the quality of the identified studies, 4. transfer evidence into the health information, and 5. evaluate the information during or after development. The questionnaire used quite detailed questions and also explored each of the five criteria using two to five sub-questions, however leaving room for the participant to answer in free text format. The particular questions are presented in Table 2. The questionnaire was first piloted with two developers and revised accordingly after discussing upcoming issues in the research group. As from our previous experiences with assessing health information material we were aware that some of the development processes would not fulfil many of the quality criteria, nevertheless, the challenge was to make administering the questionnaire a meaningful endeavour for all participants. To avoid frustration, it was necessary to form the core questions using a couple of easier questions, which were not supposed to be analysed. In addition, it was important to ensure that all developers regardless of the level of quality of their health information would at least understand the questions. Developers were assured, that they would not be contacted later on but were encouraged to contact us e.g., to request information about the guideline and the ongoing validation study. Two students from the Medical University of Graz, blinded with regard to the information gathered with the online questionnaire, rated the pool of health information material using MAPPinfo. In case the materials provided information about more than one decision (e.g., information on diagnostics, treatment and rehabilitation of the same disease), the respective parts of the materials were to be rated separately, implying a potential sample size of N = 50.

**Analysis.** Questionnaires collected via LimeSurvey were analysed by two members of the research group (NP, JK). Insights provided by the developers were used to appraise five additional MAPPinfo items (MAPPinfo-plus, see Table 2). These MAPPinfo-plus cover the

**Table 2. Appraisal of additional quality criteria (MAPPinfo-plus).**

| |
|---|
| **MAPPinfo-plus 1:** Endpoints reported in the information material are reflecting the needs of the target group. |
| **Corresponding Lime-survey questions:** Choices made about subject and contents<br>• How did the topic come about?<br>• What is the importance of a health information on the topic?<br>• Have measures been employed previous to the development (e.g., literature search or own studies), to investigate special needs and preferences of the target group regarding the relevant contents? Please describe.<br>• How does the concrete development reflect those preferences (e.g., selection of endpoints to report from studies used in the health information)? |
| **MAPPinfo-plus 2:** Evidence collected by a systematic literature search was taken as the starting point in the development of the information. |
| **Corresponding Lime-survey questions:** Identification of the evidence<br>• Please describe the steps of gaining information on the contents of the health information.<br>• Was the content given in the health information obtained through systematic literature searches?<br>• What in particular was searched for (search strategy)?<br>• Where did you search?<br>• How did you search? Please justify choices made about the proceedings involved in selecting information. |
| **MAPPinfo-plus 3:** Decisions about which evidence to use for informing the content of the health information were made based on structured quality assessment of the studies. |
| **Corresponding Lime-survey questions:** Consideration of the quality of the evidence<br>• How was the quality of the studies appraised?<br>• Have validated checklists been used?<br>• How were the quality assessments used in the development and/or presentation of the health information?<br>• Please justify choices made about the proceedings involved in selecting information. |
| **MAPPinfo-plus 4:** Transfer of data from the chosen studies into the health information followed a standardised procedure. |
| **Corresponding Lime-survey questions:** Quality assurance of the transfer of contents into the health information<br>• Please describe the procedures used for transferring contents from the studies into the health information.<br>• Was a standard procedure used (e.g., data extraction template / prepared standardised result tables)?<br>• Has a procedure been used for assuring correctness of the contents in the health information? Please provide details. |
| **MAPPinfo-plus 5:** The health information was evaluated using appropriate methods. |
| **Corresponding Lime-survey questions:** Status of evaluation of the health information<br>• What has been done to gather feedback on the quality of the health information?<br>• Was the target group involved in the evaluation during the development process?<br>• Has a systematic data collection been conducted during user testing?<br>• If applicable, please provide details about the group of persons used for testing the health information.<br>• Please describe what has been assessed and by which methods.<br>• Has an expert review been employed? And how was the person chosen?<br>• Please provide detail about how evaluation results have led to a revision of the health information. |

aforementioned quality criteria in the guideline, which refer to methods used during the development process of health information material. They were designed analogous to the 19 MAPPinfo items of the checklist and had a dichotomous answering format:

Two mean scores were calculated: MAPPinfo-mean is based on the 19 items of the checklist, however, using items 12 to 15 with a double weight as they belong to both categories, content and presentation. Arithmetically, MAPPinfo-mean is therefore based on up to 23 items. The total number of items can vary dependent on the type of information coded (one item is relevant for diagnostic information only) and on whether graphical formats are used to present frequencies in a health information. MAPPinfo-plus is calculated also as a mean score, however, based on the extended questionnaire, including the 23 items of the checklist and the five additional items (28 in total). MAPPinfo-plus is referred to as the reference standard in this study. It comprises the most comprehensible database on adherence of a health information with the guideline recommendations. A Pearson's or - in case the two scores were not normally distributed–Spearman's correlation coefficient was used to determine the extent to which MAPPinfo is estimating the reference standard accurately, reflecting construct validity

of the screening instrument. To assess the percentage of variance of the MAPPinfo-plus score which is explained by the MAPPinfo score, we fitted a linear regression model.

Table 2 shows the five additional MAPPinfo criteria and the questions used in the Lime survey to collect the information. Rating of MAPPinfo-plus, was done dichotomously (by NP and JK) based on a manual. MAPPinfo-plus1 was approved, if it became evident that choices made on the content of the health information were justified by either own studies or literature on patient needs in this condition. MAPPinfo-plus2 was approved, if it was made credible that a clear search strategy had been applied. MAPPinfo-plus3 was approved, if studies had been quality appraised and a strategy for selection of studies based on grading was apparent. MAPPinfo-plus4 was approved, if a systematic procedure of data extraction had been used. MAPPinfo-plus5 was approved, if three conditions of an appropriate evaluation were evident, e.g., results of an evaluation had been used to better the health information.

## Substudy 5: Comparison with an alternative instrument

To establish an estimate of divergent validity, MAPPinfo evaluations were compared with results from applying EQIP (EQIP 36 Items [10]) which is another frequently used instrument in the field. This was done in a cross-sectional design attached to substudy 4.

**Sample.** The study used the same test sample of 50 health information materials as in substudy 4. Two additional students from the Medical University of Graz were recruited for the EQIP ratings. MAPPinfo ratings had already been done in substudy 4.

**Data collection.** Raters applying the EQIP were blinded with regard to the information gathered with the online questionnaire and MAPPinfo. EQIP comprises 36 items in three domains (18 content, 6 identification and 12 structure). The domain identification corresponds with transparency in MAPPinfo, content with content and structure with presentation. Although similar in structure, the two instruments differ with regard to the way the criteria are operationalised and the extent the criteria are evidence-based.

**Analysis.** MAPPinfo mean scores and domain scores as well as EQIP mean scores and domain scores were calculated and associated in a correlation matrix using Spearman's correlation coefficients. As within the MAPPinfo checklist, evidence-based criteria are allocated predominantly in the domain presentation, we expected divergence between the two instruments to be documented between the EQIP scale "structure" and the MAPPinfo/presentation.

## Results

### Content validity and feasibility

The feedback of two experts on the first version of the MAPPinfo checklist was quite supportive with regard to idea, form, operationalisation of the criteria, exhaustiveness and distinctiveness of the item set. The experts provided suggestions for relatively small changes in the explanations given in the manual to define the criteria. Based on these comments, the research group consented a first revision.

Two additional revisions were conducted based on results from piloting the application of the checklist by health and nursing science students. The latter also predominantly referred to refining the wording in the general part of the manual and for some of the single items in the manual. Differentiation between the answering categories was considered unclear and difficult to define in several items. To allow for sufficient reliability, the answering format was, therefore, simplified from trichotomous to dichotomous in some cases. Apart from those sporadic occasions the checklist was perceived as well understandable, consistent and usable without any training.

## Reliability

The pretest was conducted three times with changing rater groups (in total 6 raters). Most of the items showed good inter-rater-reliability immediately and the general judgement by the expert team was sufficiently correlated with the consensus rating between the two raters. However, it was found difficult to rate four items reliably and these were therefore adapted several times until sufficient agreement was achieved for all items (Table 3). This process implied repeating the sequence of substudy 2 for the respective items. The final inter-rater-reliability was strong in average (mean of T = .79); item related T-values ranged from .52 to 1.0. Spearman's correlation coefficient between expert judgement and MAPPinfo ratings was satisfying with .61 (p< = .001).

## Construct validity

In total we needed to contact 149 developers of 139 different health information materials (39 from Austria, 86 from Germany and 14 from Switzerland) to identify 75 eligible developers, who were provided with the link to the online questionnaire. 51 developers participated and answered the questionnaire. Some of the health information materials chosen by the developers were more complex and addressed several problems related to a certain condition, such as prostate cancer, information on PSA screening, treatment and rehabilitation. If applicable, the same developer would then be counted for participation two or three times with regard to different separable parts of the health information. Finally, we collected developer-based background information from 57 health information materials (15 (26.3%) on diagnostics or screening; 35 (61.4%) on treatment; 7 (12.3%) on prevention). The developers were health insurances (n = 12), confessional bodies (n = 1), patient organisations (n = 29), doctors' surgeries (n = 23), public organisations (n = 20), hospital trusts (10), specialist organisations and foundations (20) and commercial providers (34).

MAPPinfo and MAPPinfo-plus were similarly quite low (MAPPinfo-mean = .302, SD = .176; MAPPinfo-plus-mean = .305, SD = .181) by 30% criteria compliance in average. Empirically, the scores ranged from 5% respectively 7% to 85% respectively 88%. Average scores on the level of single criteria are shown in Fig 1.

The regression analysis revealed a very strong association between the MAPPinfo-plus score and the MAPPinfo score, B = 0.98; 95%CI [0.89–1.06]. In total, 90.5% of the variation in the MAPPinfo-plus score was explained with the items of the checklist. Given the limited sample size, confidence intervals for B were constructed using bootstrapping with 1000 repetitions. According to the Pearson correlation coefficient, both scores were highly correlated (.951, p< = .001). These results indicate the satisfying predictive power of the screening instrument with regard to the reference standard additionally including appraisal of the methods used during the development process. As the more comprehensive and sound quality appraisal by the reference standard (MAPPinfo-plus) is accurately predicted by the screening instrument, MAPPinfo, construct validity is considered good.

## Divergent validity

The mean total score of the EQIP based appraisals was 0.67 (theoretical range: 0 to 1, empirical range: 0.43 to 0.90; categories: content: 0.58 (0.28–0.94), identification: 0.6 (0.17–0.83), structure: 0.8 (0.6–1).

The total scores of MAPPinfo and EQIP were moderately associated (Spearman's 0.69). The same applied to the category pairs: EQIP content & MAPPinfo definitions/content (Spearman's 0.46/ 0.56) and EQIP identification & MAPPinfo transparency (Spearman's 0.63). As hypothesised, EQIP structure & MAPPinfo presentation were poorly correlated (Spearman's 0.13).

**Table 3. Results on inter-rater-reliability.**

| Item | Topic | 1. Pretest | | | 2. Pretest | | | 3. Pretest | |
|---|---|---|---|---|---|---|---|---|---|
| | | T | PA | Revision | T | PA | Revision | T | PA |
| 1 D | Condition | .36 | 57 | Answering format | .36 | 32 | Wording, answering format | .52 | 76 |
| 2 D | Goal | .33 | 55 | Manual | .17 | 22 | Item wording | .65 | 76 |
| 3 T | Authorship | .34 | 59 | Manual | .71 | 80 | ➜ | .71 | 80 |
| 4 T | Financing | .76 | 88 | | .64 | 82 | ➜ | .64 | 82 |
| 5 T | Conflict | .96 | 94 | | .88 | 94 | ➜ | .88 | 94 |
| 6 T | Updated | .78 | 84 | | .77 | 84 | ➜ | .77 | 84 |
| 7 T | Sources | .85 | 90 | | .68 | 78 | ➜ | .68 | 78 |
| 8 T | Search | 1 | 100 | | .98 | 98 | ➜ | .98 | 98 |
| 9 C | Problem | .27 | 63 | Manual | .36 | 68 | None | .76 | 88 |
| 10 C | Options | .17 | 45 | Manual | .17 | 22 | Manual | .56 | 70 |
| 11 C | Uncertainty | .88 | 94 | | .92 | 96 | ➜ | .92 | 96 |
| 12 C | Prevalence | .88 | 94 | | .68 | 84 | ➜ | .68 | 84 |
| 13 C | Benefit | .76 | 88 | | .84 | 92 | ➜ | .84 | 92 |
| 14 C | Harms | .80 | 90 | | .92 | 96 | ➜ | .92 | 96 |
| 15 C | Testquality | 1 | 100 | | 1 | 100 | ➜ | 1.0 | 100 |
| 16 P | Language | .20 | 47 | Answering format | .64 | 82 | ➜ | .64 | 82 |
| 17 P | Narratives | .80 | 90 | Manual | .92 | 96 | ➜ | .92 | 96 |
| 18 P | Diagrams | 1 | 100 | | .92 | 94 | ➜ | .92 | 94 |
| 19 P | Framing | 1 | 100 | | 1 | 100 | ➜ | 1 | 100 |
| | Average | | | | | | | .79 | 88 |

The table shows results of inter-rater-reliability and percentage agreement (PA) over three rounds of pretesting the checklist. The table also indicates, where revisions were made to improve the reliability. Criteria with satisfying reliability in the second test ➜ (were not tested again in the third pretest).

## Discussion

Following a previously published protocol [9] the new checklist for quality of health information, MAPPinfo, has in the current study passed a comprehensive agenda of subsequent studies which together should determine its properties with particular regard to inter-rater reliability and validity.

The instrument was considered by experts to be convincing regarding operationalisation of the existing guideline [7] and easy for untrained health science students to understand and use. After some minor adjustments all items reached at least moderate, however, on average high inter-rater-reliability. Importantly, the screening of the easily accessible criteria revealed a very good estimate of the entirety of quality criteria making it justifiable to omit the demanding assessment process of the reference standard. Finally, the comparison of MAPPinfo with the EQIP instrument [10] underlined the uniqueness of the new instrument and revealed detailed indication of similarity and diversity in a plausible manner. In particular, diversity between the instruments was seen in the category "presentation" comprising evidence-based criteria. According to the previous systematic review [9] the lack of rigorously evidence-based criteria has been considered a deficiency among many existing instruments also to be found in the EQIP.

The instrument itself and the studies conducted for its validation have some limitations.

The checklist is applicable only to information dealing with problems related to decisions implying that if there are no options to choose between, not even an alternative such as waiting before a treatment or other measure is conducted MAPPinfo would not be applicable. Also,

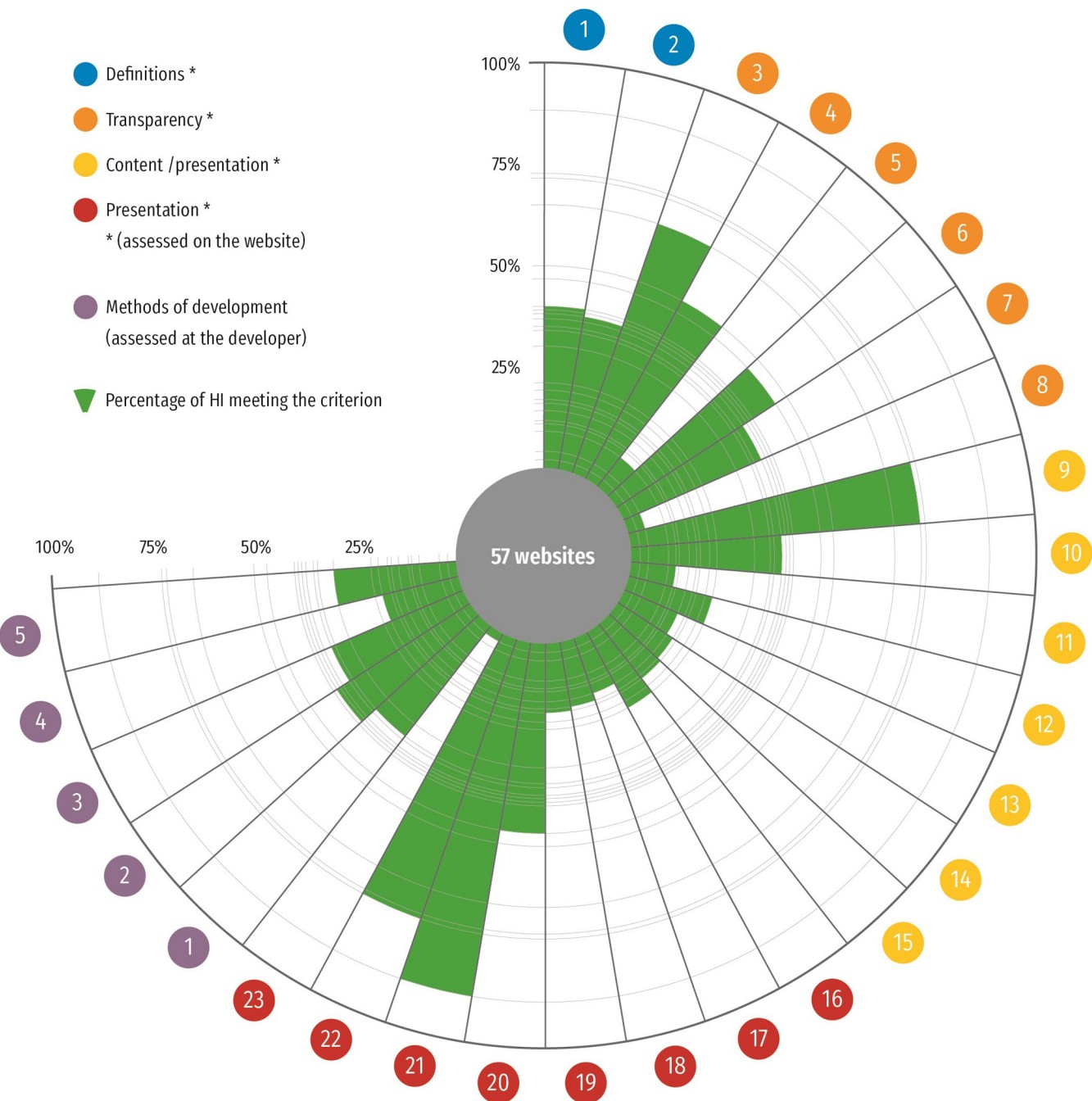

**Fig 1. Quality of the information within the construct validation sample.** The figure visualises the findings of quality for the 57 information materials used in substudy 4. Green is used to indicate compliance with respective criteria averaged over the construct validation sample. Categories marked with a "*" are included in the checklist. The figure comprises all items (1–23) of the MAPPinfo checklist and the additional items employed in the construct validation (substudy 4, items 1–5 in purple).

information on topics, such as on the structure of the health system or information providing guidance through a procedure or a treatment after a decision has been made, are not assessible with the checklist. However, we assume that informing health decisions is the crucial genre of

health information in general, and there are few decisions not including the alternative of not doing anything.

Moreover, we have to bear in mind that by exclusively relying on the ethically determined and the clearly evidence-based criteria, MAPPinfo lacks many other potentially important features in its assessment. This implies that the quality appraisal refers to the parts of the landscape which have already been mapped through research evidence. For example, the way photos of people are used in health information material might have a strong impact on the quality of the product. Unfortunately, we do not yet have evidence to justify recommendations accordingly. The latter limitation also implies the need of frequent updates of both the guideline and the instrument.

From a test theoretic perspective, the checklist might be featured unfavourably as item difficulty is low and there is little graduation in the answering format of the items. Most of them are even dichotomous. This makes it difficult to accredit attempts made by the developers to comply with particular items. In the checklist, appraisal of the criteria is guided rigorously considering the perspective of the user needing them to be fully met. For example, if the benefit is appropriately presented for one of the alternative treatments but not for the others, the information is as useless for the user as it would be without an adequate presentation of the benefits.

For pragmatic reasons, the current study used self-declaration of quality assurance steps by the developers as a reference standard in the validation design. Due to the developers' potential tendency to answer in accordance with what they assumed would be viewed as desirable by the researchers, this decision might have led to some bias. The gold standard would have meant, in addition to collecting all this information from the developers, to more or less repeat the entire development process of the information. However, this was not practicable. The correlation is very high, nevertheless.

Finally, the method used for determining construct validation is limited by the fact that the score of the checklist is part of the reference standard, implying interdependence. Although its strength is not surprising or even likely with regard to the numerical ratio of criteria (5/28) the association (90% explanation of variance) is conclusive, showing that the easily accessible criteria reveal a sufficiently precise estimate of the more complex and comprehensive approach. The informative value of this prediction does, however, not go beyond its practical implications, such as the advantage that quality assessment is now transparent for the public and feasible to whom it might concern. A complete construct validation of the new instrument would, in addition, use a third measure, e.g., informed choices made by the users to compare the predictive power of both instruments. We would consider such an approach worthwhile, but not urgent, as according to the guideline all criteria have already proven effective with regard to the facilitation of informed choices.

In recent years, health literacy and shared decision-making seem to be one of the most emphasised topics of research and debate within the literature on health communication. Both approaches try to optimise the users' prerequisites of making soundly informed health choices. The situation, however, reminds us of the emperor presenting his new clothes to a frenetic crowd [16]. Both the emperor and the crowd do not dare to recognize the obvious: The clothes are missing. In the same way, the core element of supporting users in making informed choices–health information of sufficient quality - seems to have been largely neglected [17–23].

Health literacy has repeatedly [24] been found to be far from sufficient in the general population and very low amongst risk groups and groups with a migration background. The good news is that the latter findings might not tell much about actual health literacy as the competences assessed have not proved to be health promoting Moreover, the findings are based on self-assessment and can be interpreted as self-efficacy that is related to health information

rather than as a competence [25]. However, uncertainty about what these huge surveys really mean also implies that we do have even worse insight into the level of people's health literacy. Even highest levels of health literacy will not allow for the application of the latter competence in a concrete situation when an individual is challenged by a health problem as long as material informing the particular decision does not exist. This implies a crucial role of both the availability of health information on as many problems as possible and the compliance with the already documented quality criteria for the people's health. Healthcare providers cannot compensate for lacking or insufficient information. They might have access to scientific literature and medical guidelines, but to be able to provide information to the patient they would themselves need this information to be prepared according to the recommendations of the guideline evidence-based health information [7].

The new checklist is the first of its kind providing the opportunity to map the actual quality of existing information. Therefore, the instrument has been chosen as an outcome for a current trial evaluating a training intervention in compliance with the guideline evidence-based health information [23]. As the instrument is concise, well-structured and easily accessible it might also stimulate compliance with scientific guidelines and even be used to certify health information developments.

The checklist is currently being used to map parts of the health information landscape in three European countries (Austria, Germany and Norway). This is considered important research, determining the actual capacity of health information to facilitate informed choices. Results will inform policymakers in their attempts to define effective measures to strengthen people's health literacy.

## Conclusions

The new checklist, MAPPinfo, has proven reliable and valid. It is ready for use and freely available on the homepage of "Stiftung Gesundheitswissen" [26]. It is the first providing the opportunity to scale up transparent, effective and nevertheless evidence-based quality assessment of health information. The concept of quality operationalised by MAPPinfo is well justified but limited to criteria with a clear ethical or scientific evidence. More research on design features of health information is urgently needed.

## Acknowledgments

We would like to thank all the developers who participated in the construct validation study for their active contribution and their willingness to share details of the development process. We are also grateful for the work done by health and nursing science students as raters applying the new instrument. Also, we have to thank Jan Keppler, who participated in recruiting developers and as a rater and Martina Albrecht and Susanne Buhse for providing expert reviews on the draft of the instrument. Moreover, we wish to thank the Stiftung Gesundheitswissen for inspiring cooperation on the project, constructive and encouraging discourse as well as financial contribution.

## Author Contributions

**Conceptualization:** Jürgen Kasper, Julia Lühnen, Jana Hinneburg, Andrea Siebenhofer, Nicole Posch, Birte Berger-Höger, Anke Steckelberg.

**Data curation:** Jürgen Kasper, Nicole Posch, Anke Steckelberg.

**Formal analysis:** Jürgen Kasper, Nicole Posch, Alexander Grafe, Milada Cvancarova Småstuen, Anke Steckelberg.

**Funding acquisition:** Jürgen Kasper, Andrea Siebenhofer, Nicole Posch, Anke Steckelberg.

**Investigation:** Jürgen Kasper, Nicole Posch, Anke Steckelberg.

**Methodology:** Jürgen Kasper, Julia Lühnen, Jana Hinneburg, Andrea Siebenhofer, Nicole Posch, Birte Berger-Höger, Alexander Grafe, Milada Cvancarova Småstuen, Anke Steckelberg.

**Project administration:** Jürgen Kasper, Nicole Posch, Anke Steckelberg.

**Resources:** Jürgen Kasper, Andrea Siebenhofer, Nicole Posch, Anke Steckelberg.

**Software:** Jürgen Kasper, Nicole Posch, Anke Steckelberg.

**Supervision:** Jürgen Kasper, Andrea Siebenhofer, Nicole Posch, Anke Steckelberg.

**Validation:** Jürgen Kasper, Jana Hinneburg, Andrea Siebenhofer, Nicole Posch, Anke Steckelberg.

**Visualization:** Jürgen Kasper, Nicole Posch, Anke Steckelberg.

**Writing – original draft:** Jürgen Kasper, Julia Lühnen, Jana Hinneburg, Andrea Siebenhofer, Nicole Posch, Birte Berger-Höger, Alexander Grafe, Milada Cvancarova Småstuen, Anke 'Steckelberg.

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
