## [Decision Letter · Decision Letter 0]

25 May 2023

PONE-D-22-33017MAPPinfo - mapping quality of health information: validation study of an assessment instrumentPLOS ONE

Dear Dr. Kasper,

Thank you for submitting your manuscript to PLOS ONE. After careful consideration, we feel that it has merit but does not fully meet PLOS ONE’s publication criteria as it currently stands. Therefore, we invite you to submit a revised version of the manuscript that addresses the points raised during the review process.

Please follow the instructions of the two reviewers (1 and 3) who have asked for some changes to be made to the paper, not only with respect to the contents of the different sections (in particular, method, results and discussions), but also with respect to the figures and tables, thus increasing the readability and clarity of the paper's contents

We look forward to receiving your revised manuscript.

Kind regards,

Ramona Bongelli, Ph.D.

Academic Editor

PLOS ONE

Journal Requirements:

Additional Editor Comments:

Dear authors,

my sincere apologies for the revision time, which took longer than I expected.

As you can see, two of the three reviewers who responded to our invitation are asking you to make changes to the submitted paper.

Reviewers' comments:

Reviewer's Responses to Questions

**Comments to the Author**

1. Is the manuscript technically sound, and do the data support the conclusions?

Reviewer #1: Yes

Reviewer #2: Yes

Reviewer #3: Partly

2. Has the statistical analysis been performed appropriately and rigorously? 

Reviewer #1: Yes

Reviewer #2: Yes

Reviewer #3: Yes

3. Have the authors made all data underlying the findings in their manuscript fully available?

Reviewer #1: No

Reviewer #2: Yes

Reviewer #3: Yes

4. Is the manuscript presented in an intelligible fashion and written in standard English?

Reviewer #1: Yes

Reviewer #2: Yes

Reviewer #3: No

5. Review Comments to the Author

Reviewer #1: The article presents an assessment tool that is the result of valuable, rigorous and interesting studies, both in terms of scope and future potential, considering the major shortcomings in the field of health information and thus the participatory deficit in the decision-making process.

I note some points for review:

1) In general, there is a need to make the paper more discursive and usable, including providing some more specific information on some protocols, guidelines, criteria, cited as references (e.g., EQIP, CREATE, EBHI, 19 MAPPinfo items) since the initial framing (even simply explaining them in the notes). At present, the article is very, sometimes too much, schematic, both in the divisions into parts and in the explanations of individual parts.

2) In particular, it is suggested to expand and deepen:

- the abstract, as much as possible in the limited space

- discussion/conclusions, especially regarding what can be done to overcome the limitations of MAPPinfo, the practical application of the tool, future developments in health information

3) Fig. 1 is not easy to understand. The items thus presented (23+5) and the percentage values do not help to understand the results that are intended to be shown.

The items should be listed either in the text or put in a legend.

4) Check that all data collected and used are entered as the Plos Data policy requires.

5) In substudy 5: the sentence "EQIP comprises 36 items in four categories (18 content, 6 identification and 12 structure)" is incongruent? four categories but three are listed...

6) Is MAPPinfo based on 19 criteria (as stated on p.1) or 19 items as on p. 7?

7) It would be interesting to also attach the list of developers of health informations involved and see if a particular category is more likely to provide satisfactory health information

8) Bibliography should be expanded a bit.

With this minor revision, the article is deemed publishable.

Reviewer #2: A very important paper with high quality. However I think, that the abstract should be revised, I think it does not reflect the high quality of the paper, especially the method and the results should be better described.

Reviewer #3: Thank you for the opportunity to review the manuscript “MAPPinfo - mapping quality of health information: validation study of an assessment instrument”. The manuscript describes the validation of the a newly developed instrument that allows users to assess the quality of health information. Qualitative and quantitative methods are used to test the interrater reliability, comprehensibility and convergent validity of the instrument is tested. Developing a validated instrument to support the quality assessment of health information is an important research endeavor. However, I have a number of observations that should be addressed. In the following I provide more detailed comments. I hope they will be helpful in developing this interesting line of research.

1. Detailed description of the systematic review. In the background section you mention that you conducted a systematic review. Please provide more information about the procedure and results of the systematic review.

2. The background section would benefit from a more detailed description of the instrument. Although the fact sheet provides general information, I suggest a more detailed description of the categories and the process by which the specific items were developed. As you compare the MAPPinfo with an existing instrument (the EQIP), you should also explain the added value of the MAPPInfo. What does it offer that existing instruments, such as the EQIP do not? The section should also explain the differences between MAPPInfo and MAPPInfo-Plus and the rationale behind both scores/instruments. This information is currently missing. However, without this information the results section is difficult to follow. In addition, information should be provided on the users for whom the instrument was developed. From the introduction, I had the impression that the user group was lay people, but in the description of study 2, health and nursing students are mentioned as target group. Please clarify this. Finally, please provide more information and arguments for the different validation steps. Why have these steps been taken and what information with regard to the validity of the instrument are they intended to provide?

3. In the subsection “Aim of the study”, you argue that you compared the quality assessment based on MAPPinfo with a quality assessment based on an established instrument (i.e., EQIP) in order to test divergent validity. As I understand it, divergent validity helps to establish construct validity by showing that the construct of interest is different from other constructs. Convergent validity, on the other hand, provides information on whether scores obtained from different measures of the same construct converge. As you have used different measurement instruments to assess the quality of health information, you have tested convergent rather than divergent validity.

4. Description and adequacy of the CREATE-guidelines: As not all readers may be familiar with the CREATE guidelines, please provide a more detailed description of the guidelines. In addition, you should explain, why the CREATE guidelines were applied to your evaluation study. The CREATE guidelines were developed for multi-attribute utility-based instruments and I am not sure whether the MAPPinfo is indeed such an instrument and whether the guidelines are appropriate for the validation process. Regardless of this criticism, much of the information described in the CREATE guidelines is not provided in the manuscript. Therefore, if you do decide to use this guideline, please follow the recommendations when presenting your research.

5. As detailed in the CREATE guidelines, a rationale for the sample size selected should be provided for all studies. For study 2, it is unclear how many people participated in the validation study. In the first round of the feasibility test two students participated. After revisions, new versions of the instrument were tested, but no information is provided on how many iterative revision rounds were conducted and how many people participated in these rounds. Finally, there is no description of the criteria used to decide that no further revisions were necessary.

6. Study 3: Please provide information on how the students rated the quality of the health information. Did they use the same rating scale from very low quality to excellent quality as the experts? Please change “Likert scale” into “rating scale” as Likert scales measure dis/agreement and not quality. Please provide more information about the number of rounds of data collection and the final number of study participants.

7. Study 4: The section on the analysis is difficult to follow. Please provide more detailed information on what kind of analyses were conducted and how they contribute to establishing the validity of the instrument. For example, what is meant by the MAPPinfo-plus score and the MAPPinfo score? What does the regression model tell us? Please also explain the five items listed in the manuscript in more detail. As described in the manuscript, missing answers were interpreted as negative answers. However, this is not an appropriate way of dealing with missing data. I suggest that missing data be excluded from the analyses. Furthermore, information should be provided on how many answers were missing per item. Finally, please explain how the answers to the questionnaires were combined to form the MAPPinfo score and the MAPPinfo-plus score, respectively?

8. Study 5: What does the sentence “This has been done in a cross-sectional design attached to study 4” mean? As the same sample was used as in study 4, study 5 is not an independent study but should be described together with study 4.

9. Results: Please provide a table with the information on interrater-reliability for all items. From a validation point of view, it is not appropriate to establish convergent validity by comparing the newly developed instrument with itself (MAPPinfo and MAPPinfo-plus). I had the impression that two versions of the instrument were tested to show whether they are equivalent. To test convergent validity, the scores of the newly developed measure must be compared with the scores of the EQIP. The results section must be modified accordingly. Moreover, the results on convergent validity using the EQIP are difficult to follow. Therefore, I was not able to judge the appropriateness of the analyses and the interpretations.

10. The discussion should be more explicit about the value of the Instrument and what it adds to existing literature.

11. Overall, the writing should be improved. At present, much information is missing or described in a way that it is difficult to follow and understand.

6. PLOS authors have the option to publish the peer review history of their article (what does this mean?). If published, this will include your full peer review and any attached files.

Reviewer #1: No

Reviewer #2: No

Reviewer #3: No

---

## [Author Response · Author response to Decision Letter 0]

23 Jun 2023

see point by point response letter submitted as separate document

---

## [Decision Letter · Decision Letter 1]

25 Jul 2023

PONE-D-22-33017R1MAPPinfo - mapping quality of health information: validation study of an assessment instrumentPLOS ONE

Dear Dr. Kasper,

Thank you for submitting your manuscript to PLOS ONE. After careful consideration, we feel that it has merit but does not fully meet PLOS ONE’s publication criteria as it currently stands. Therefore, we invite you to submit a revised version of the manuscript that addresses the points raised during the review process.

I would ask you to follow the very last small revision requests below

We look forward to receiving your revised manuscript.

Kind regards,

Ramona Bongelli, Ph.D.

Academic Editor

PLOS ONE

Journal Requirements:

Reviewers' comments:

Reviewer's Responses to Questions

**Comments to the Author**

1. If the authors have adequately addressed your comments raised in a previous round of review and you feel that this manuscript is now acceptable for publication, you may indicate that here to bypass the “Comments to the Author” section, enter your conflict of interest statement in the “Confidential to Editor” section, and submit your "Accept" recommendation.

Reviewer #3: (No Response)

2. Is the manuscript technically sound, and do the data support the conclusions?

Reviewer #3: Yes

3. Has the statistical analysis been performed appropriately and rigorously? 

Reviewer #3: Yes

4. Have the authors made all data underlying the findings in their manuscript fully available?

Reviewer #3: (No Response)

5. Is the manuscript presented in an intelligible fashion and written in standard English?

Reviewer #3: Yes

6. Review Comments to the Author

Reviewer #3: Thank you for your revisions. I offer the following suggestions:

Abstract: lines 34-37 The terms criteria and items are not used consistently.

Abstract lines 58 onwards. The sentence “By definition, …” needs elaboration.

Lines 66-67: It is unclear which information and which decision-making processes the authors refer to. Please add this information.

Line 95: Please explain the acronym MAPPinfo.

Line 226: I suggest changing “. A Pearson‘s or - in case the two scores could not be normally distributed” to – in cases the two scores were not…

Lines 321, having experience in scale development, I still doubt that divergent validity is the adequate term for the analysis. To show divergent validity, correlation coefficients should be low rather than as high as .69 for the total scores.

I still suggest a professional proofreading to improve the readability of the paper.

7. PLOS authors have the option to publish the peer review history of their article (what does this mean?). If published, this will include your full peer review and any attached files.

Reviewer #3: No

---

## [Author Response · Author response to Decision Letter 1]

27 Jul 2023

Dear Editor, (a point by point response letter has been uploaded as a separate doc - might be easier to read.

Please receive our grateful thanks for additional comments from the third reviewer on our first revision. We have revised the manuscript according to this feedback. Please find below a point by point summary of all changes. Changes in the main manuscript are marked in blue.. 

Best regards, 

Jürgen Kasper on behalf of the author team

Reviewers' comments:

Reviewer #3: Thank you for your revisions. I offer the following suggestions:

1. Abstract: lines 34-37 The terms criteria and items are not used consistently.

Response: We agree and have rephrased the sentence: 

Therefore, we drafted the checklist, MAPPinfo, addressing the existing criteria with 19 items.

2. Abstract lines 58 onwards. The sentence “By definition, …” needs elaboration.

Response: Thank you, we agree. The sentence is difficult because of its double negation. We have rephrased the sentence.

According to its underpinning concept the instrument exclusively comprises items which are justified by either ethics or research evidence, implying negligence of not yet evidence based, however, potentially important criteria.

3. Lines 66-67: It is unclear which information and which decision-making processes the authors refer to. Please add this information.

Response: Thank you for this comment. Since your reference are the lines 66-67 which are the empty lines under the abstract, we understand the comment as generally related to the abstract. And we agree. We, therefore, added a definition of informed choices: 

 – decisions, made by individuals about their own health based on knowledge and in congruence with own preferences.

4. Line 95: Please explain the acronym MAPPinfo.

Response: Thank you, we added the full name of the project in the last paragraph of the background. 

5. Line 226: I suggest changing “. A Pearson‘s or - in case the two scores could not be normally distributed” to – in cases the two scores were not…

Response: Thank you, we changed the sentence accordingly.

6. Lines 321, having experience in scale development, I still doubt that divergent validity is the adequate term for the analysis. To show divergent validity, correlation coefficients should be low rather than as high as .69 for the total scores.

Response: We fully agree with your definition of the divergent vs. convergent validity. And our intention was to demonstrate divergence in order to justify the need of the new instrument. We also agree with your appraisal of a correlation of .69 as demonstrating more convergence rather than divergence. So maybe we must be become better in admitting that EQiP is not given enough diverging results. However, our approach in this substudy was more focussing on the pattern of subscales than on the total scale. We are not surprised to see similar results in scales measuring corresponding criteria. But we expected to disclose divergence when it comes to evidence based criteria like the mode of presentation. 

This rationale is expressed in the methods section by: As within the MAPPinfo checklist, evidence-based criteria are allocated predominantly in the domain presentation, we expected divergence between the two instruments to be documented between the EQIP scale “structure” and the MAPPinfo/presentation.

7. I still suggest a professional proofreading to improve the readability of the paper.

Response: We have thoroughly read through the manuscript another time and improved the language at several places. The manuscript has previously been edited two times by a professional translator. Therefore, we believe the English is appropriate.

---

## [Editor Report · Decision Letter 2]

1 Aug 2023

MAPPinfo - mapping quality of health information: validation study of an assessment instrument

PONE-D-22-33017R2

Dear Dr. Kasper,

We’re pleased to inform you that your manuscript has been judged scientifically suitable for publication and will be formally accepted for publication once it meets all outstanding technical requirements.

Kind regards,

Ramona Bongelli, Ph.D.

Academic Editor

PLOS ONE
---

## [Editor Report · Acceptance letter]

22 Aug 2023

PONE-D-22-33017R2 

MAPPinfo - mapping quality of health information: validation study of an assessment instrument 

Dear Dr. Kasper:

I'm pleased to inform you that your manuscript has been deemed suitable for publication in PLOS ONE. Congratulations! Your manuscript is now with our production department. 

Kind regards, 

on behalf of

Professor Ramona Bongelli 

Academic Editor

PLOS ONE